# COVID-19 Vaccine-Associated Ocular Adverse Effects: An Overview

**DOI:** 10.3390/vaccines10111879

**Published:** 2022-11-07

**Authors:** Parul Ichhpujani, Uday Pratap Singh Parmar, Siddharth Duggal, Suresh Kumar

**Affiliations:** Department of Ophthalmology, Government Medical College and Hospital, Sector-32, Chandigarh 160030, India

**Keywords:** COVID-19 vaccine, booster, molecular mimicry, ocular adverse effects

## Abstract

Background: To address the pandemic caused by severe acute respiratory syndrome coronavirus 2 (SARS-CoV-2), vaccination efforts were initiated across the globe in December 2020 and are continuing. We report the onset interval and clinical presentations of ocular adverse effects following SARS-CoV-2 vaccination. Methods: For this narrative review, articles in the English language, published between 1 January 2020 to 1 September 2022, were included to formulate a list of the reported ocular adverse effects of different COVID-19 vaccines. Results: During this period, ocular adverse effects have been reported with BNT162b2 (Pfizer), mRNA-1273 (Moderna), AZD-1222 (AstraZeneca), and Ad26.COV2.S (Johnson & Johnson) vaccines. Endothelial graft rejection, herpes simplex virus keratitis, herpes zoster ophthalmicus, anterior uveitis, eyelid edema, purpuric rashes, ischemic optic neuropathy, and cranial nerve palsies were the most reported with BNT163b2. Retinal hemorrhages, vascular occlusions, and angle closure glaucoma were the most reported with AZD-1222. Most of the ocular adverse effects reported in the literature had a good to fair prognosis with appropriate management. Conclusions: Evidence regarding the ocular adverse effects does not outweigh the benefits of SARS-CoV-2 vaccination in patients with pre-existing systemic or ophthalmic diseases. This review provides insights into the possible temporal association between reported ocular adverse events and SARS-CoV-2 vaccines; however, further investigations are required to identify the link between potential causality and pathological mechanisms.

## 1. Introduction:

The coronaviruses are positive sense, single-stranded ribonucleic acid (RNA), enveloped medium-sized viruses. Coronaviruses are classified as a family within the order Nidovirales with a spike (S) glycoprotein which mediates receptor binding and cell entry. S protein is the site of the major antigens that stimulate neutralizing antibodies and target cytotoxic lymphocytes, thus making it an important vaccine antigen. Seven different strains of coronaviruses that infect humans include the common cold coronavirus strains; 229E, NL63, OC43, and HKU1 and the more pathogenic strains include Middle East respiratory syndrome (MERS)-CoV, severe acute respiratory syndrome (SARS)-CoV, and SARS-CoV-2. Since the structure and function of pathogenic strains of coronaviruses causing diseases like SARS and MERS were known it helped in the early development of various vaccine platforms across the globe. 

A stepwise approach for developing any new vaccine involves vaccine development, clinical trials, U.S. Food and Drug Administration (FDA) approval or authorization, manufacturing, and distribution. The COVID-19 vaccines were developed at an unprecedented pace and were given Emergency Use Authorizations (EUAs) [1]. As of 19 September 2022, a total of 12,640,866,343 vaccine doses have been administered. COVID-19 vaccines and updated/bivalent COVID-19 boosters are effective at protecting people from being hospitalized, serious illness, and death [2]. 

Currently, 11 COVID-19 vaccines have been approved for EUA, which can be subdivided into four types: mRNA vaccines (BNT162b2, Pfizer-BioNTech14; mRNA-1273, Moderna15), protein subunit vaccines (NVX-CoV2373, Novavax16), vector vaccines (Ad26.COV2, Janssen Johnson & Johnson17; ChAdOx1 nCoV-19/AZD1222, Oxford-AstraZeneca18), and whole virus vaccines (PiCoVacc, Sinovac19; BBIBP-CorV, Sinopharm20) (Table 1) [3]. Individual vaccine trials report vaccine safety with rare ocular adverse effects but given the massive scale of the current vaccination drive, the possible adverse effects are a cause for concern. Since the widespread administration of COVID-19 vaccinations, multiple reports of ocular adverse effects after COVID-19 vaccinations and boosters have emerged [4,5]. 

To develop methods for closely observing ‘at risk’ patients, reporting of adverse effects must be conducted on a regular basis. This narrative review summarizes ocular adverse effects that are possibly associated with COVID-19 vaccination. The aim is to encourage early recognition of adverse effects not only by ophthalmologists but also by treating physicians. 

## 2. Methodology

A literature search was performed in PubMed for ‘COVID-19 vaccine’, ‘ocular inflammation’, ‘ophthalmic manifestations’, ‘adverse effects’, ‘graft failure’, ‘retinal hemorrhage’, ‘uveitis’, ‘neuro-ophthalmology’, ‘nerve palsy’, and ‘vascular occlusion’. Articles of interest were searched using Boolean operators. Each synonymous word was separated by a Boolean operator, “OR”, phrases were enclosed within quotation marks, and groups of synonymous words were enclosed within parenthesis. Articles in the English language, published between 1 January 2020 to 1 September 2022, were included to formulate the list of the reported ocular adverse effects of different COVID-19 vaccines. The search, although not exhaustive, includes important and relevant articles. Search results were screened by two authors (PI and SD) for relevance. References cited within the identified articles were also used to further augment the search. We characterized our results into an anterior segment, posterior segment, and neurophthalmic adverse effects.

## 3. Results

Ocular complications reported post-COVID-19 vaccination included abducens nerve palsy, oculomotor nerve palsy, facial nerve palsy/Bell’s palsy, multiple cranial nerve palsies, acute macular neuroretinopathy (AMN), paracentral acute middle maculopathy (PAMM), superior ophthalmic vein thrombosis (SOVT), corneal graft rejection, anterior uveitis, panuveitis, central serous chorioretinopathy(CSCR), Vogt–Koyanagi–Harada (VKH) reactivation, acute zonal occult outer retinopathy (AZOOR) and multifocal choroiditis. The reported ocular adverse effects following vaccination appear to overlap with the ocular manifestations of COVID-19 itself, suggesting a common pathway between virus- and vaccine-mediated immune response in humans. Aggregated information on the reviewed cases is elucidated in Table 2 and Table 3.

## 4. Discussion

A new variant of CoV emerged in Wuhan, China in December 2019 that caused severe respiratory illness. The World Health Organization named this virus SARS-CoV-2 and the pandemic COVID-19. According to Li, Y.-D. [55], to address the global morbidity and mortality caused by COVID-19, the development process of COVID-19 vaccines was expedited by undertaking clinical trials in parallel rather than in a linear fashion. Multiple COVID-19 vaccines directly entered clinical trials on humans without preclinical testing in animal models. The COVID-19 vaccination drive has been carried out worldwide and the evidence is overwhelming that irrespective of the type(s) of vaccine taken, the vaccines offered safety and protection against becoming seriously ill or dying due to the different variants of CoV-2.

The Vaccine Adverse Event Reporting System (VAERS) was developed by the U.S. Food and Drug Administration (FDA) in 1990 as a national early monitoring system for vaccine safety. The commonly reported adverse effects of COVID-19 vaccinations consist of the injection site’s local reaction followed by several non-specific flu-like symptoms. However, several systemic and organ-specific (e.g., eye, heart) adverse effects have also been reported from across the globe. Therefore, it is imperative for ophthalmic health care providers to be familiar with the clinical presentations, pathophysiology, diagnostic criteria, and management of ocular adverse effects following COVID-19 vaccination. Early diagnosis and quick initiation of the treatment may help to provide patients with a more favorable outcome and rule out masquerading entities. With an increasing amount of literature in the form of isolated case-study reports, case series, and analysis of the VAERS database, an epidemiological montage has started to emerge [56].

A recent Lancet article questioned the effectiveness of COVID-19 vaccines and the waning of immunity over time, more pronounced in individuals with pre-existing conditions and elderly adults. According to Nordström, P. [57], in addition to the risk of infections owing to lowered immune function, the authors cited a possible risk of some organ damage caused by the vaccine that has remained somewhat sequestered in the circulatory system, without apparent clinical presentations. This can explain the slightly delayed presentation of some of the adverse effects.

Vaccines have added adjuvants within them to boost their efficacy; these adjuvants potentiate the innate and adaptive immune responses further, possibly leading to autoimmune or inflammatory conditions in some individuals. Although truncated and modified RNA traces may be present in BNT162b2 and mRNA-1273 vaccines, these aberrant proteins have a minuscule chance of eliciting allergic reactions. The active constituent of the vaccine is not always the culprit for causing adverse reactions. Excipients such as polyethylene glycol (PEG) used in the BNT162b2 and mRNA-1273 vaccines have been reported to have induced IgE-mediated allergic reactions [3].

Despite reports suggesting an association between ocular adverse effects and the vaccines due to a maladaptive immune response in susceptible individuals, the adverse issues are still considered ‘rare’ given the millions of people who have received either one or more vaccines or boosters.

The COVID-19 vaccines interact with the platelets or the platelet factor 4 (PF4) and this interaction results in vaccine-induced immune thrombotic thrombocytopenia (VITT). The proposed mechanisms suggest the formation of autoantibodies against PF4, antibodies induced by the free deoxyribonucleic acid (DNA) in the vaccine that cross-reacts with PF4, platelets, and adenovirus binds to the platelets causing platelet activation. VITT may explain vascular occlusions [58].

Endothelial graft rejection, herpes simplex virus (HSV) keratitis, herpes zoster ophthalmicus (HZO), anterior uveitis, eyelid edema, purpuric rashes, ischemic optic neuropathy, and cranial nerve palsies were the most reported with the BNT163b2 vaccine. Although both BNT162b2 and mRNA 1273 are mRNA vaccines, the ocular adverse effects have been relatively lesser with mRNA 1273 than those with BNT162b2. 

Retinal hemorrhages (subretinal, subhyaloid, or intraretinal), vascular occlusions, and angle closure glaucoma have been the most reported with the AZD 1222 vaccine. No COVID-19 vaccine-associated adverse events have been reported in patients with age-related macular degeneration in the peer-reviewed literature to date.

The pathophysiological mechanisms underlying vaccine–corneal graft rejection are still poorly understood. However, cases of acute graft rejection have also been reported following influenza, hepatitis B, yellow fever, and tetanus toxoid vaccinations. Steinemann, T.L. and Wertheim, M.S. [59,60] proposed mechanisms for acute corneal allograft rejection include the reduction in the corneal immune privilege due to systemic immune dysregulation and activation of toll-like receptors on the ocular surface and CD4+ T helper-1 cell (Th1) immunity. Corneal edema was the leading clinical manifestation, followed by keratic precipitates in patients with corneal graft rejection. Most of the ocular adverse effects reported in the literature had a good to fair prognosis with appropriate management. Therefore, corneal graft recipients should not be discouraged from receiving COVID-19 vaccines or boosters. Additionally, the evidence is insufficient to suggest delaying keratoplasties or uptitrating topical steroid administration after a routine keratoplasty, following primary COVID vaccine or booster administration. In high-risk cases, increasing immunosuppressants in the peri-vaccination period may decrease the risk of immune reactions [9].

Studies suggest a link between COVID-19 vaccines and the reactivation of the varicella-zoster virus (VZV), resulting in vaccine-acquired immunodeficiency syndrome. Data from the works of Barda, N., Desai, H.D. and Seneff, S. [61,62,63] have shown that the population prevalence rates of post-vaccination ophthalmic HSV were ≤0.05 cases per million doses and for HZO were ≤0.5 cases per million doses. According to Wang, M.T.M. [64], there is no conclusive evidence to suggest the need for prophylactic antiviral treatment for patients with prior herpetic eye disease considering COVID-19 vaccination.

Regarding vaccine-associated uveitis (VAU), a recent VAERS review by Singh R B et al. reported a total of 1094 cases from 40 countries with an estimated crude reporting rate (per million doses) of 0.57, 0.44, and 0.35 for BNT162b2, mRNA-1273, and Ad26.COV2.S, respectively. More than two-thirds of cases were reported in patients who received BNT162b2. Additionally, the post hoc analysis showed a significantly shorter interval of onset for the first dose compared with the second dose and BNT162b2 compared with the mRNA 1273 vaccine. According to [65], other vaccines that have also triggered uveitis flare-ups include hepatitis A and B, influenza, Bacillus Calmette–Guérin, human papillomavirus, measles–mumps–rubella (MMR), and varicella zoster vaccines. According to Wang [64], De novo VKH cases can be the result of molecular mimicry between vaccine peptide fragments and uveal self-peptides, whereas, for cases with VKH reactivation, specific HLA haplotypes may account for the individual susceptibility of the autoimmune activation. Many patients who developed ocular adverse effects lacked medical comorbidities that may have predisposed them to the adverse effects, although a few patients were on hormone-based birth control [49,66].

Although clinical trials for all vaccines undergo rigorous safety monitoring prior to authorization for human use, some serious adverse events may not be identified in trials, especially if uncommon, because of the relatively small sample size, the selection of trial participants who may not represent the general population, restrictive eligibility criteria, and limited duration follow-up [67].

The data regarding ocular adverse effects with other approved vaccines, such as ZyCoV-D, Sputnik, Covidecia, Sputnik, Abdala, Zifivax, and Novavax are sparse. Despite the mandatory requirement by all nations to report any vaccine-associated adverse events, unreliable reporting, under-reporting, and/or delayed reporting are common. Additionally, the possibility of anti-vaccination fringe groups attempting to malign vaccines using VAERS data by adding misinformation about the safety of COVID-19 vaccinations must also be remembered.

To conclude, the scientific evidence regarding the ocular adverse effects does not outweigh the benefits of COVID immunization in patients with pre-existing systemic or ophthalmic conditions. However, patients must be counseled to seek prompt medical review for symptoms of post-vaccination deterioration of vision or primary ocular disease relapse.

## Figures and Tables

**Table 1 vaccines-10-01879-t001:** List of WHO-approved vaccines for COVID-19.

SNo	Name	Type of Vaccine	Country Where Vaccine Was Developed	Countries That Have Used It	Route of Admin	VVM	Preservatives	Diluents
1	Covovax (Novavax formulation)	Protein Subunit (Recombinant Nanoparticle)	India (Serum Institute of India)	6 countries	IM	N/A	N/A	N/A
2	Nuvaxovid (Novavax)	Protein Subunit	Czech Republic	40 countries	IM	N/A	N/A	N/A
3	mRNA-1273 Moderna: Spikevax	RNA (modified nucleoside)	Spain (Moderna Biotech)	88 countries	IM	N/A	N/A	N/A
4	BNT163b2 Pfizer BioNTech: Comirnaty	RNA (Modified nucleoside)	Germany (BioNTech Manufacturing GmbH)	149 countries	IM	N/A	N/A	Sodium Chloride Inj USP 0.9%
5	Convidecia: CanSino (Ad5.CoV2-S)	Non replicating viral vector	People’s Republic of China (CanSino Biologics Inc.)	10 countries	IM	None	None	None
6	Jcovden: Janssen (Johnson & Johnson)	Non-replicating viral vector	Belgium (JCINV)	113 countries	IM	None	None	None
7	Vaxzevria (Oxford AstraZeneca)	Non replicating viral vector	Republic of Korea (AstraZeneca/SK Bioscience Co., Ltd.)	149 countries	IM	None	None	None
8	Covidshield (ChAdOx1 nCoV-19 (AZD1222) (Oxford AstraZeneca formulation)	Non-replicating viral vector	India (Serum Institute of India)	49 countries	IM	None	None	None
9	Covaxin	Inactivated (Whole virion)	India (Bharat Biotech)	14 countries	IM	N/A	Phenoxy ethanol	N/A
10	Sinopharm: Covilo/BBIBP-CorV	Inactivated (Antigen is purified and absorbed with aluminium hydroxide)	China (BIBP)	93 countries	IM	VVM7	None	N/A
11	Sinovac: Coronavac	Inactivated (Antigen is purified and absorbed with aluminium hydroxide)	China (Sinovac Biotech)	56 countries	IM	N/A	None	N/A

(IM: Intramuscular; RNA: Ribonucleic acid; VVM: Vaccine Vial Monitor Type).

**Table 2 vaccines-10-01879-t002:** Anterior segment manifestations following COVID-19 vaccines.

Manifestation		Vaccine	Time of Onset	Symptoms	Case/Case Series Age/Age Range	Mechanism	Treatment and Outcome	Article Reference No.
Endothelial Graft Rejection	1	BNT163b2	7 days 3 weeks	Painless decrease in vision Red eye	66 yrs 83 yrs Case report	Allogenic response, generated by the host antibodies and immune system	Treated successfully with topical steroids	(Phylactou, M et al., 2021) [6]
	2	BNT163b2	7 days	Sudden painless decrease in vision, conjunctival injection; diffuse corneal edema	71 yrs Case report	Disruption of immune regulation and upregulation of cytokines like TNF α, chemokines, and pro inflammatory molecules	Treated with topical Dexamethasone sodium phosphate 1 mg/mL/2 hourly Resolution after 2 weeks	(Crnej, A et al., 2021) [7]
	3	BNT163b2	14 days	Painless worsening of vision Corneal thickness increased, OCT Descemet membrane folds	94 yrs Case report	Changes in antibody-mediated immune signalling response following vaccination	Dexamethasone/tobramycin With hypertonic saline	(Forshaw, T et al., 2022) [8]
	4	BNT163b2 (8 patients)	17 days 3 weeks × 2 13 days 14 days 7 days 3 days 4 days 4 days 9 days 13 days	Conjunctival hyperemia, diffuse corneal edema, KPs, flare and cells, corneal thickness, stromal edema reported in 1 patient	Systematic review Median age 68 (27–83) IQR	Increased anti-spike-neutralizing antibodies, antigen-specific CD4^+^ T-cell responses, and inflammatory cytokines, including interferon (IFN)-γ and interleukin-2 IFN-γ plays a central role in the acute rejection process and the resultant T helper type 1-dominant immune response may have evoked corneal allograft rejection	Dexamethasone eye drops 0.2% hourly, combined oral methyl prednisone, hypertonic saline, intracameral fortecortin injections	(Fujio, K et al., 2022) [9]
		mRNA-1273 (8 patients)	1 week 1 week 2 week 1 week 15 days 3 days 1 week 1 week					
		ChAdOx1 (4 patients)	5 days 10 days 2 days 6 weeks					
		CoronaVac	1 day		63 yrs			
	5	BNT163b2 (3 cases)	16.86 ± 6.96 days (mean)	Painless loss of VA and conjunctival suffusion	Case series	Hyperstimulation of the immune system	Topical Steroids	(Molero-Senosiain, M et al., 2022) [10]
		AZD1222 (2 cases)	17 ± 11.89 days					
	6	BNT163b2	2 weeks	Decreased VA, ocular pain, photophobia	73 yrs Case report		Prednisone acetate every 1–2 h, with Muro ointment	(Abousy, M et al., 2021) [11]
	7	CoronaVac Biotech	24 h		63 yrs Case report		Partially resolved by topical corticosteroids and polydimethylsiloxane	(Simão, M.F et al., 2022) [12]
	8	ChAdOx1 COVIDSHIELD, AstraZeneca	2 weeks	Blurring of vision, stromal edema	28 yrs Case report		Hourly topical steroids, cycloplegics and oral steroids	(Nahata, H et al., 2022) [13]
	9	mRNA-1273 (4 cases)	3 weeks 9 days 2 weeks 2 weeks		Case report		Topical steroids Complete resolution	(Shah, A.P et al., 2022) [14]
Herpes zoster Ophthalmicus (HZO)	1	mRNA-1273	6 days	Itchy tender lesions on the right thigh, eruption of vesicles with an erythematous base	79 yrs Case report	Lymphopenia along with any functional impairment of T lymphocytes could trigger herpes zoster reactivation	Complete resolution after systemic antiviral treatment	(Eid, E et al., 2021) [15]
	2	BNT163b2 (Tozinameran) (2 cases)	15 days 13 days	Painful grouped vesicles in the left lateral of the ox coccyges (S3 dermatome) Painful and swollen inguinal lymph nodes along with a rash on the right leg	29 yrs 34 yrs Case report		Self-limiting Valacyclovir 1 g 3×/day for 10 days, complete resolution	(van Dam, C.S et al., 2021) [16]
	3	AZD-1222 (Covidshield)	4 days	Multiple grouped fluid filled lesions on an erythematous base, present on the knee and the anterior aspect of the thigh; biopsy showed acantholytic cells and dyskeratotic cells	60 yrs Case report		Valacyclovir 1 g 3×/day for 7 days Topical Fusidic acid 2×/day	(Arora, P et al., 2021) [17]
	4	mRNA-1273 (14 cases)	2, 0, 4, 4, 14, 12, 2, 0, 12, 12, 26, 5, 4, 5–6 days, respectively	Unilateral dermatological skin eruptions, with itching, pain, arm soreness, altered skin sensation	77, 56, 54, 69, 42, 47, 39, 68, 60, 43, 65, 37, 69, 72 yrs, respectively	Immunomodulation due to decrease in lymphocytes, monocytes, eosinophils, CD4/CD8 T cells	Valacyclovir Gabapentin LMX, Terrasil Shingles cream	(Lee, C et al., 2021) [18]
		BNT163b2 (6 cases)	38, 5, 3, 12, 9, 5 days, respectively		65, 43, 74, 48, 46, 44 yrs, respectively Case series			
	5	BNT163b2 (5 cases)	1, 5, 3, 2, 16 days, respectively	Umbilicated vesicles, lymphadenopathy, dysesthesias, fever, vesicles and rash in dermatomal pattern	58, 47, 39, 56, 41 yrs, respectively Case series		Valacyclovir 1 g 3×/day for 7 days	(Rodríguez-Jiménez, P et al., 2021) [19]
	6	RNA vaccine	5 days	Painful pimple-like lesions with stinging in the left mammary region, crusted haemorrhagic vesicles upon an erythematous base	78 yrs Case report		Valacyclovir 3×/day for 7 days	(Bostan, E et al., 2021) [20]
	7	Inactivated COVID 19 vaccine	5 days	Multiple pinhead vesicular lesions with an erythematous base occupying right mammary region and back along with stinging sensation and pain	68 yrs Case report		Valacyclovir 3×/day for 7 days Codeine for pain management	(Aksu, S.B et al., 2021) [21]
	8	BNT163b2 (7 cases)	9, 14, 8, 7, 9, 7, 10	Unilateral dermatomal rash in different dermatomes (lumbar, thoracic, 5th cranial nerve) malaise, headache	51, 56, 89, 86, 90, 91, 94 yrs Case series		Valacyclovir for 7 days after symptoms onset	(Psichogiou, M et al., 2021) [22]
Herpes Simplex Virus (HSV) Keratitis	1	ChAdOx1n	1 day	Corneal hyperaemia, reduced corneal sensation, multiple corneal dendrites, reduced VA (6/9)	82 yrs Case report	Molecular mimicry, autoinflammation triggered by the vaccine and lymphopenia	Acyclovir 5×/day, doxycycline 50 mg orally once a day, prednisone phosphate 0.5% Atropine 1% 1×/day	(Richardson-May et al., 2021) [23]
	2	Sinovac (2 cases)	2 days 4 days	Tearing, redness, photophobia, decreased visual acuity, dendritic lesions on slit lamp examination	60 yrs 51 yrs Case report	Lymphopenia, insufficient cellular immunity	Topical steroids, topical and oral ganciclovir	(Rallis et al., 2022) [24]
	3	BNT163b2	2 days	Blurry vision, 20/40 VA, patchy stromal haze and confluent punctate epithelial erosions along inferior cornea	52 yrs	Activation of the proinflammatory cytokines like INF gamma post vaccination, can have a role in reactivation	Acyclovir 5×/day Topical trifluride 5×/day Prednisolone acetate 1% 5×/day	(Fard et al., 2022) [25]
		mRNA-1273	2 weeks	Blurry vision, redness, abnormal sensation in OS, corneal epithelial defect	67 yrs		Bandage contact lens, Oral Valacyclovir 1 gm 2×/day Ofloxacin 0.3% drops 4×/day	
	4	BNT163b2 (3 cases)	1 week 1 week 1 week	Pain, Photophobia, Lacrimation Typical dendritic ulcer of the peripheral cornea Stromal infiltration and diffuse conjunctival injection AC trace	18 yrs 40 yrs 29 yrs	Vaccine triggered cytokine release and upregulation of natural killer cells group D ligand, causing reactivation	Lubrication, ganciclovir ophthalmic gel 0.15% 5×/day Oral acyclovir 400 mg 5×/day for 10 days	(Alkwikbi, H et al., 2022) [26]
		AZD1222	1 week	Pain, redness, blurry vision; epithelial dendritic ulcers were noted on the cornea	32 yrs Case report		Prednisone 1 mg/kg/day for 4 weeks	
	5	BNT163b2 (2 cases)	4 days 4 weeks	Necrotizing stromal keratitis Endothelitis and epithelial keratitis	42 yrs 29 yrs Case report	Potential immunological dysregulation	Systemic acyclovir	(Alkhalifah, M.I et al., 2021) [27]
	6	BNT163b2	5 days	Conjunctival hyperaemia, pseudodendrite in peripheral cornea Along with vesicular skin rash on forehead, scalp, nose, eyelid, meningitis	74 yrs Case report	Temporal association due to immunological upregulation of cellular immunity	Therapeutic contact lens, recombinant human epithelial growth factor, Ofloxacin ointment	(You et al., 2022) [28]
	7	BNT163b2	2 days	OS redness, tearing, and pain	50 yrs Case report	mRNA vaccines dysregulate T cell latency mechanisms in the sensory nerve ganglion	400 mg oral acyclovir + Topical fluorometholone	(Al-Dwairi et al., 2022) [29]
	8	ChAdOx1n	7 days	Pain, photophobia, blurred vision, Examination—peri corneal injection, hazy cornea with paracentral thinning	56 yrs Case report	Potential immune response triggered by molecular mimicry	Topical and systemic acyclovir	(Murgova et al., 2022) [30]
		BNT163b2 (4 cases)	3 weeks 8 days 2 weeks 10 days	Blurred vision and irritation in OS, KPs seen in the anterior segment Moderate vitritis and exudates	89 yrs 50 yrs 52 yrs 45 yrs		Acyclovir Steroids (1 mg/kg methylprednisone)	
	9	BNT163b2	2 days	Sudden visual impairment, diffuse corneal stromal edema, nasal stromal infiltration	87 yrs Case report	T cells activation by the host cell response after vaccination may have caused the recurrance	Oral Valacyclovir and topical corticosteroids	(Ryu et al., 2022) [31]
	10	BNT163b2	7 days	Decreased VA and foreign body sensation	30 yrs Case report	Unknown immunological response or general systemic reactogenicity to the vaccine-causing reactivation	Topical eye drops (Ganciclovir 0.25%) Loteprednol etabonate 0.5%	(Song et al., 2022) [32]
Anterior Uveitis	1	BNT163b2 (3 cases)	6 days 6 days 8 days	Photophobia Blurred vision	44 yrs 47 yrs 44 yrs	Immunological hyperstimulation by the vaccine	Dexamethasone eye drops 2 mg/mL leading to complete resolution	(Bolletta et al., 2022) [33]
		AZD1222	30 days	Redness, pain, blurred vision	66 yrs			
		mRNA-1273	1 day	Redness, pain, blurred vision	35 yrs Case report			
	2	BNT163b2	14 days	Pain, photophobia and red eye. Conjunctival hyperaemia, posterior synechiae and AC cells, KPs in the lower quadrants	23 yrs	Molecular mimicry	10-day course of topical steroids and cycloplegics	(Renisi et al., 2022) [34]
	3	Sinopharm	5 days	Reduced VA, hyperreflective dots in the AC, fine endothelial granularities	18 yrs Case report	Potential immunological mechanisms	Topical steroids leading to complete resolution	(ElSheikh et al., 2021) [35]
	4	BNT163b2	3 weeks	Acute onset pain, photophobia, erythema, blurring of vision	46 yrs Case report	Autoantibodies production post vaccination as a component of the hyper-stimulated immune system reacting with self peptides	Topical triamcinolone drops, azathioprine 50mg once daily as a steroid-sparing agent	(Al-Allaf et al., 2022) [36]
	5	BNT163b2	3 days	Redness, blurred vision, headache Corneal epithelial edema, KPs in the lower quadrant	54 yrs Case report	Secondary molecular mimicry due to similarity between the vaccine fragments and the peptides of the uvea, adjuvants such as aluminium cause inflammatory damage, delayed hypersensitivity response	0.1% Dexamethasone, 1% Cycloplegic drugs, 0.1% dexamethasone ointment	(Duran 2022) [37]
	6	BNT163b2	2 months	Photophobia, redness, decreased vision, pain Intense ciliary flush and posterior synechiae	37 yrs Case report	mRNA vaccine-induced cellular and humoral immune responses, which can lead to molecular mimicry and immunological cross-reactivity	Topical prednisone acetate 1% and cyclopentolate	(Alhamazani et al., 2022) [38]
	7	BNT163b2	3 days 3 days	Ocular pain, redness, hemicranial headache	92 yrs 85 yrs Case report	Molecular mimicry and antigen specific cell and antibody-mediated hypersensitivity reactions	Cycloplegic every 8 h and moxifloxacin eye drops every 4 h	(Ortiz Egea et al., 2022) [39]
	8	BNT163b2	2 days	Decreased VA and conjunctival injection Hypopyon and flares in the AC	21 yrs Case report	Vaccine-induced molecular mimicry	Topical dexamethasone (0.1%) hourly and systemic prednisone (50 mg/day) for 7 days	(Hwang JH. 2022) [40]
	9	AZD-1222	1 day	Greater vitreous opacity, KPs, increase in inflammatory cells in the AC	62 yrs	Molecular mimicry between vaccine and ocular structures leading to autoreactivity	Topical steroids	(Choi et al., 2022) [41]
		BNT163b2 (2 cases)	3 days 2 days		79 yrs 55 yrs		Topical steroids	
Episcleritis and Anterior Scleritis	1	AZD1222		Anterior non-necrotising scleritis				(Hernanz I et al., 2022) [42]
	2	Sinopharm (3 cases)	1 week 15 days 15 days	Diffuse scleral hyperemia	33 to 55 yrs Case series	Molecular mimicry and antigen-specific cell and antibody-mediated hypersensitivity reactions	Resolved in 2 weeks after topical steroids	(Pichi F et al., 2022) [43]
Angle closure glaucoma	1	AZD-1222	2 weeks	Ocular pain, acute visual loss, corneal microscopic cystic edema, conjunctival injection, shallow AC, peripheral AC collapse	71 yrs	Swelling of ciliary body after vaccination, that led to zonule laxity accompanied by phacodonesis, causing a closed-angle attack	Phacoemulsification with goniosynechiolysis	(Choi M et al., 2022) [41]
	2	AZD-1222			83 yrs		Trabeculectomy with laser peripheral iridoplasty	
	3	AZD-1222			59 yrs		Phacoemulsification with posterior chamber lens implantation	
	4	AZD-1222			64 yrs		Vitrectomy with IOL scleral fixation	
	5	BNT163b2	3 days	Blurry vision, headache, corneal edema,	54 yrs	Not mentioned	20% Mannitol, acetazolamide 250 mg, timolol, dorzolamide, 0.15% brimonidine	(Duran M 2022) [37]
Eyelid edema	1	BNT162b2	1 day	Transient eyelid edema	39.3 mean age (32–43) Case series	Complement activation that increased complement mediators within the plasma and tear film, resulting in eyelid edema	Observation, Antihistamine, Corticosteroid	(Austria QM et al., 2021) [44]
Purpuric eyelid rash	1	BNT162b2	Median of 18 days	Purpuric rashes on the upper lids associated with mild itching	44 yrs 63 yrs 67 yrs	Mild and localized form of vaccine-induced microangiopathy	Self-resolving	(Mazzatenta et al., 2021) [45]
Bell’s palsy	1	BNT162b2	3 days	Latero-cervical pain in left side, irradiating to the mastoid ipsilaterally, monolateral muscle weakness; flattening of the forehead skin and nasolabial fold	37 yrs	Possible autoimmune reaction	Corticosteroids (prednisone, 50 mg/day), artificial tears eye drops and eye dressing at night Helped resolve systemic symptoms, facial mobility partially improved and pain sensation still persists	(Colella et al., 2022) [46]

(AC: anterior chamber; KP: keratic precipitates; OCT: optical coherence tomography, OS: oculus sinistrum; VA: visual acuity).

**Table 3 vaccines-10-01879-t003:** Posterior segment and neurophthalmic manifestations following COVID-19 vaccines.

Manifestation	Vaccine	Time of Onset (Days)	Symptoms	Case/Case Series Age/Age Range	Mechanism	Visual Prognosis	Article Reference No.
Vitreo-Retina:							
Acute Macular Neuroretinopathy (AMN)	AZD1222	3 days	Bilateral paracentral scotomas with underlying bilateral circumscribed paracentral dark lesions on ophthalmoscopy, OCT with outer plexiform layer thickening and discontinuity	21 yrs	Hypoperfusion of the retina might account for the peripheral visual loss, which self-corrected rapidly. Reduction in central acuity is less straight forward to explain, but can result from transient hypoperfusion of retina, optic nerves, or any part of the visual pathways extending to the visual cortices. [7]	Self-limiting	(Book et al., 2021) [47]
	AZD1222	2 days	Unilateral paracentral scotoma with a teardrop-shaped macular lesion nasal to the fovea	27 yrs		Symptoms only lasted for 24 h	(Mambretti et al., 2022) [48]
	AZD1222	2 days	Unilateral presentation with paracentral scotoma	22 yrs		Self-limiting	(Mambretti et al., 2021) [48]
	AZD1222	2 days	Unilateral presentation with paracentral scotoma	28 yrs			(Mambretti et al., 2021) [48]
	BBIBP-CorV Sinopharm	5.2 days (range, 1–10 days)	Previous ocular history of CSCR in both eyes with a chronic serous PED in the OS. BCVA of 20/25 at previous visits. Vital parameters were within normal limits, but the BCVA in OS dropped to 20/400.	41.4 (9.3) yrs (range: 30–55 yrs)	Can be associated with anemia, hypertension or hypotension, hypoxia, and other systemic morbidities, which can contribute to nerve fiber layer infarcts, haemorrhages, or microaneurysms. Vasculitis and thromboembolism also can contribute to retinal ischemia.	Patient was closely observed, and at 2-month follow- up, the tomographic picture had resolved and BCVA was back to 20/30.	(Pichi et al., 2021) [43]
2. Paracentral acute middle maculopathy (PAMM)	AZD1222 2nd dose	30 days	Reduced brightness sensitivity in both eyes progressed further, black spots in his central field. OCT macula revealed a significant reduction in the number and size of the hyperreflective lesions noted in the nerve fibre and ganglion cell layers. There was also a reduction in the thickness of the outer nuclear layer in both eyes.	35 yrs	Possible microvascular pathology affecting the deep capillary plexus. It is possible that small vessel vasculitis induced by vaccination resulted in these findings. It is hypothesized that the vasculitis changes may have led to the ischemia of the deep capillary plexus presenting as PAMM and AMN in the patient.	On re-examining the patient after 3 weeks he reported slight improvement of brightness sensitivity but still complained of black spots in his central field of vision. On examination, his vision was 6/6 in both eyes. His Amsler’s grid charting was also normal. The OCT macula revealed a significant reduction in the number and size of the hyperreflective lesions noted in the NFL and GCL. There was also a reduction in the thickness of the outer nuclear layer in OU.	(Vinzamuri et al., 2021) [49]
	BBIBP-CorV Sinopharm	5.2 days (1–10 days)	20 min after receiving Sinopharm, they developed persistent tachycardia and raised blood pressure. Noticed inferior scotoma in OS. BCVA at presentation was 20/30 OS, with a dot hemorrhage superior to the fovea.	41.4 (9.3) yrs (30–55 yrs)	Molecular mimicry and antigen-specific cell and antibody-mediated hypersensitivity reactions may be involved.	BP was nonresponsive to treatment for 3 weeks	(Pichi et al., 2021) [43]
3. Multifocal choroiditis	AZD1222	7 days	Patient had a large unilateral serous macular detachment and severe choroidal thickening bilaterally. BCVA was 6/36, N60, and 6/6, N6 in OD and OS, respectively.	34 yrs	Autoimmunity triggered by the vaccines. Mechanisms include cytokine production, expression of human histocompatibility leukocyte antigens, modification of surface antigens, induction of novel antigens, molecular mimicry, bystander activation, epitope spreading, polyclonal activation of B cells, and an immune reaction to vaccination adjuvants known collectively as Shoenfeld syndrome are often associated with constitutional symptoms such as arthralgia, myalgia, and fatigue.	On oral prednisolone 100 mg daily (1 mg/kg body weight) tapering by 10 mg/week after 11 days the patient reported significant improvement in vision. UCVA improved to 6/6, N6. Significant resolution of choroiditis with trace residual subretinal fluid. B-scan showed significant reduction in CT.	(Goyal et al., 2021) [50]
4. Panuveitis	BNT162b2	3 days	Patient presented with a visual acuity of 20/500 in both eyes, eye pain, eye redness, and sensitivity to light having 3–4+ anterior chamber cell with 2–3+ vitreous cell with significant choroidal thickening.	43 yrs	Direct infection of ocular structures by live strain (the COVID-19 vaccine is not a live strain * Additive-induced immune-related uveitis (which are not present in the Pfizer-Biontech vaccine) * Molecular mimicry between the vaccine and ocular structures, driving the adaptive immune system to create autoimmunity.	Within 10 days of starting oral prednisone pain resolved, her VA improved to 20/20 OU, there was no inflammation, and the choroidal thickening resolved.	(Mudie et al., 2021) [51]
5. Central serous chorioretinopathy (CSCR)	BNT162b2	3 days	Unilateral blurry vision. BCVA OD: 20/63; OS: 20/25 with metamorphopsia	33 yrs	Possibly due to increased serum cortisol, free extracellular mRNA, and polyethlene glycol.	At the 2-month visit, BCVA:20/40; CFT: 325 μm. At the 3-month visit, BCVA: 20/20; CFT: 211 μm. OCT showed complete resolution of subretinal fluid	(Fowler at al., 2021) [52]
6. Acute Zonal Occult Outer Retinopathy (AZOOR)	mRNA-1273	10 days	Bilateral presentation with progressive unilateral nasal defect and bilateral flashes. At presentation, had 20/20 vision in both eyes with a yellow-white reflex in the temporal macula of her left eye.	33 yrs	Possible mechanisms include: (i) molecular mimicry, where the vaccine triggers an immune response to self-antigens; (ii) bystander activation of sequestered self-antigens from the host that can activate antigen-presenting cells and T-helper cells; (iii) cytokines secretion from macrophages that recruit additional T-helper cells; (iv) genetic polymorphisms related to the aberrant regulation of the IL-4 expression or activity, which may over-stimulate inflammatory responses	Patient was recommended a combination therapy of azathioprine and cyclosporine. Patient consulted her gynecologist prior to starting therapy as she was nursing a baby.	(Maleki et al., 2021.) [53]
7. Posterior Uveitis	BNT162b2 (4 cases) AZD1222 (4 cases) mRNA-1273 (1 case)	6.5 [1,2,3,4,5,6,7,8,9,10,11,12,13,14] after first dose 8 [2,3,4,5,6,7,8,9] after second dose	Unilateral in 8, bilateral in 1. 2 (22.2%) had history of ocular toxoplasmosis, 1 (11.1%) of AZOOR. Patients with history of ocular toxoplasmosis presented with recurrence of lesions and the patient with AZOOR had a different presentation from previous events with multifocal choroiditis. 3 (33.3%) presented with ocular toxoplasmosis, 2 (22.2%) presented with retinal vasculitis, and 1 (11.1%) presented with choroiditis for the first time.	40 yrs	Out of 3 with previous history of posterior uveitis, 2 had history of previous similar event. * Molecular mimicry secondary to resemblance between uveal peptides and vaccine peptide fragments. * Antigen-specific cell and antibody-mediated hypersensitivit. Reactions * Inflammatory damage induced by adjuvants included the vaccines stimulating innate immunity through endosolic or cytoplasmic nucleic acid receptors.	VA unaffected in 7 (77.8%) VA reduced > 3 lines in 2 (22.2%) Macular Scarring in 2 (22.2%) On being treated by topical corticosteroid in 6 and systemic corticosteroid in 3. One patient with ocular toxoplasmosis, and 1 with occlusive retinal vasculitis had persistent vision loss on the last follow-up due to macular scarring.	(Testi et al., 2021) [54]

(BCVA: Best-corrected visual acuity; CSCR: central serous chorioretinopathy; GCL: ganglion cell layer; mRNA: messenger ribonucleic acid; NFL: nerve fiber layer; OS: oculus sinistrum; PED: pigment epithelial detachment).

## Data Availability

Not applicable.

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
