# Peer review of "COVID-19 Vaccine-Associated Ocular Adverse Effects: An Overview"

_vaccines, 2022, doi:10.3390/vaccines10111879_

Round 1

Reviewer 1 Report

Dear authors

why didn’t you consider statistical analysis of the aggregated data to find percentage of adverse events regarding each type of vaccine.     Your efforts are a great contribution to provide vigilance and find correlation between each vaccine category and potential adverse effects.

Author Response

Thank you for your time and effort for reviewing our manuscript.

We didn’t consider statistical analysis as it was a narrative review and not a metanalysis. We don’t deny the possibility of missing out some literature pertaining the ocular adverse effects. Secondly, VAERS database has its own set of limitations, and many nations don’t even have adverse effect reporting databases, thus the figure that would have emerged from the aggregated data would not have been a correct reflection of the correlation between each vaccine category and the adverse effects.

But point well taken, in a subsequent review, we will consider the most common adverse effects and do a metanalysis and report statistical data.

Reviewer 2 Report

The manuscript entitled “COVID-19 VACCINE ASSOCIATED OCULAR ADVERSE EFFECTS: AN OVERVIEW” is a review work summarizing the ocular adverse events associated post COVID-19 vaccination. My comments are mentioned below

1.      Abstract: “During this period, ocular adverse effects have been reported BNT162b2, mRNA-1273, AZD-1222, and Ad26.COV2.S, respectively”. Revise this sentence.

2.      Abstract needs to be corrected for grammatical errors.

3.      Introduction is written well, however, authors should mention the full form of the used abbreviation below the table for clarity.

4.      Discussion: The first paragraph should be in the introduction part. It will be nice to transfer this paragraph to starting paragraph of the introduction.

5.      The whole manuscript should be revised for typos and grammatical errors.

6.      The result and discussion section need restructuring.

Author Response

Reviewer 2: The manuscript entitled “COVID-19 VACCINE ASSOCIATED OCULAR ADVERSE EFFECTS: AN OVERVIEW” is a review work summarizing the ocular adverse events associated post COVID-19 vaccination. My comments are mentioned below

Abstract: “During this period, ocular adverse effects have been reported BNT162b2, mRNA-1273, AZD-1222, and Ad26.COV2.S, respectively”. Revise this sentence.

Response: Thank you for pointing out the error. Corrected.

Abstract needs to be corrected for grammatical errors.

Response: Thank you for pointing out the error. Corrected.

Introduction is written well, however, authors should mention the full form of the used abbreviation below the table for clarity.

Response: Thank you for pointing out the error. Corrected.

Discussion: The first paragraph should be in the introduction part. It will be nice to transfer this paragraph to starting paragraph of the introduction.

Response: Thank you for the suggestion. Corrected.

The whole manuscript should be revised for typos and grammatical errors.

Response: Thank you for the suggestion. Corrected.

The result and discussion section need restructuring.

Response: Thank you for the suggestion. Corrected to some extent.